# OpenReview forum: "On Evaluating the Durability of Safeguards for Open-Weight LLMs"
_ICLR.cc/2025/Conference — ICLR 2025 Poster_

### Official Review · Reviewer_nnB1 · 2024-10-18

**Soundness:** 3
**Presentation:** 3
**Contribution:** 2
**Rating:** 6
**Confidence:** 3

**Summary:**

This paper explores the complexity of evaluating the durability of security safeguards for open-weight large language models (LLMs) through detailed case studies. Focusing on two methods, RepNoise and Tamper Attack Resistance (TAR), the study reveals that even minor changes in fine-tuning configurations, dataset randomness, or prompt templates can drastically alter, or even nullify, the effectiveness of these "durable" defenses.

**Strengths:**

1. The paper offers a comprehensive analysis of challenges in evaluating safeguards for open-weight language models, showing how various factors affect their perceived effectiveness.
2. Through case studies of two defense techniques, the authors highlight key failure modes and evaluation pitfalls, revealing the limitations of current methods.
3. The insights provide valuable guidance for safety and security research, stressing the need for rigorous testing across diverse scenarios to ensure defense robustness.

**Weaknesses:**

1. The paper highlights the challenges of creating durable safeguards but does not propose new methods or solutions to address these issues, instead calling for further research, which leaves the problem unresolved.
2. The analysis is comprehensive but limited to two specific methods (RepNoise and TAR), which narrows its overall applicability.
3. The detailed exploration of how minor changes in configurations affect performance might be hard to follow for some readers.

**Questions:**

1. The paper only discusses two defense methods (RepNoise and TAR). Do other potential defenses face similar evaluation challenges?
2. Could the authors explore potential improvements to the evaluation process or provide preliminary ideas for solutions?

---

> ### Author Response · Authors · 2024-11-22
> **Rebuttal - Part I**
>
> We thank the reviewer for acknowledging the thoroughness and comprehensiveness of our study, as well as for recognizing the valuable insights it provides. We also greatly appreciate the constructive feedback for improvement. In response, we have carefully revised our work and provided clarifications below, which we hope will address the reviewer’s remaining concerns.
>
> **1. The problem of durably safeguarding open-weight LLMs is left unresolved because it is hard and still remains an open problem.**
> > The paper highlights the challenges of creating durable safeguards but does not propose new methods or solutions...
>
> We thank the reviewer for bringing up this issue. Below is our clarification on this point:
>
> * **The point of this paper is not to attempt to solve the problem but to establish baseline evaluation principles for future work that attempts to make progress.** A painful lesson we learned from AI security research over the last decade is that many defenses failed to withstand rigorous scrutiny due to flawed or incomplete evaluations (e.g., [1,2]). This highlights the importance of establishing standard baseline evaluation principles for evaluating defenses, as exemplified by [3,4]. Only if we first address the meta-problem of evaluating defenses can we genuinely track progress in building defenses and confidently draw security conclusions.
>
>     However, this evaluation issue has not received sufficient attention in recent work (such as RepNoise and TAR, which we examined in this paper) that attempts to produce durable safeguards for open-weight LLMs. These studies initially made strong claims about the effectiveness of their defenses, but upon re-evaluation, we found that they can be easily bypassed and that their evaluations contain various pitfalls, leading to overestimation of their defenses' effectiveness. This motivated us to conduct our study. By using these two defenses as case studies, we highlight common evaluation pitfalls in this area, and we hope that our findings can serve as important evaluation principles for future work attempting to make progress in this problem.
> * **Why didn't we explore alternative solutions?** The reviewer is correct that this work did not explore alternative approaches for durably safeguarding open-weight LLMs. We didn't do so because safeguarding an open-weight model when the adversary can arbitrarily modify the model's weights is a difficult open problem. The goal of our paper is not to develop a new solution, but to ensure that any exploration of solutions to this challenging problem properly evaluates the solutions and avoids pitfalls.
>
>     To appreciate its difficulty, consider that so far, we do not even have a solution for adversarial examples, where the adversary only "fine-tunes" the input to the model --- and it is already 2024, we still see researchers try very hard to solve the problem but the solution was then quickly shown to fail in openreview comments (https://openreview.net/forum?id=IHRQif8VQC).
>
>      Now, we have a much harder problem: for open-weight LLMs, how can we defend against a much stronger adversary who can directly "fine-tune" the model's weights, not just the input, as in adversarial examples? It is even unclear under what conditions durably safeguarding open-weight models is a feasible goal to pursue under such a strong threat model. All these remain open research questions to date.
>
> * **Another point of this paper is to inform practitioners of the difficulty of safeguarding open-weight LLMs with current technologies.** Not all practitioners in AI safety and security are fully aware of the actual effectiveness of current defenses. Our study provides solid evidence to help stakeholders calibrate their expectations regarding the feasibility of durably safeguarding open-weight LLMs with existing technologies.
>
>     For example, a recent public interest comment (https://www.mercatus.org/research/public-interest-comments/comment-nist/aisi-ai-800-1-expanding-guidelines-open-source-ai) suggested that the U.S. National Institute of Standards and Technology (NIST) include approaches like TAR in their guidelines for managing open-source foundation models. Our evaluation offers an evidence-based counterpoint, indicating that users and policymakers should be cautious about relying on these methods in their current state.
>
> [1] Carlini, Nicholas, and David Wagner. "Defensive distillation is not robust to adversarial examples." arXiv preprint arXiv:1607.04311 (2016).
>
> [2] Athalye, Anish, Nicholas Carlini, and David Wagner. "Obfuscated gradients give a false sense of security: Circumventing defenses to adversarial examples." International conference on machine learning. PMLR, 2018.
>
> [3] Carlini, Nicholas, et al. "On evaluating adversarial robustness." arXiv preprint arXiv:1902.06705 (2019).
>
> [4] Tramer, Florian, et al. "On adaptive attacks to adversarial example defenses." Advances in neural information processing systems 33 (2020): 1633-1645.

---

> > ### Author Response · Authors · 2024-11-22
> > **Rebuttal - Part II**
> >
> > **2. Generalizability of our study**
> >
> >
> > > The analysis is comprehensive but limited to two specific methods (RepNoise and TAR), which narrows its overall applicability.
> >
> >
> > > The paper only discusses two defense methods (RepNoise and TAR). Do other potential defenses face similar evaluation challenges?
> >
> >
> > Our experiments in this paper are to show how the security evaluations in existing work for durably safeguarding open-weight LLMs are flawed and lead to overestimation of the defenses' effectiveness. We primarily focus on RepNoise and TAR, because these were the only two papers that claim to build durable safeguards for open-weight LLMs by the time we conducted this study. Besides, we chose any of the specific models or datasets in our experiments because these defense works chose them to showcase their defenses.
> >
> >
> > Moreover, the evaluation pitfalls and principles identified in our work are not tied to any particular model or dataset—they are consolidated to be generic and widely applicable. For example, our discussion of randomness in evaluation (Section 3.1), the implementation details of fine-tuning attack trainers (Section 3.2), the adaptiveness of fine-tuning attack configurations (Section 3.3), and the choices of prompt templates and metrics (Section 3.4) apply universally, regardless of the specific model or dataset being evaluated.
> >
> > Similarly, the principles we outline for assessing whether unlearning methods genuinely remove targeted information, as well as the guidelines for reporting evaluation results and drawing security conclusions, are also model-agnostic and broadly applicable across defense evaluation scenarios (Section 4).
> >
> >
> >
> >
> > **3. Improve presentations to make readers easier to follow.**
> >
> >
> > > The detailed exploration of how minor changes in configurations affect performance might be hard to follow for some readers.
> >
> > We thank the reviewer for this suggestion. We added a new Table 3 (See L957-L962) to better introduce the hyperparameter configurations used in RepNoise. We have detailed the hyperparameter configurations used by TAR in Tables 1 and 5.
> >
> >
> >
> >
> >
> >
> > **4. A more organized summary of evaluation principles.**
> > > Could the authors explore potential improvements to the evaluation process or provide preliminary ideas for solutions?
> >
> >
> > We thank the reviewer for this suggestion. In our rebuttal revision, we added Appendix A to provide a more explicit and actionable checklist of items to consider for future evaluations of defenses to avoid similar pitfalls we identified in our paper. The checklist includes tips for:
> >
> >
> >  (1) Checking the robustness of the defense against attacks with different random seeds;
> >
> >
> >  (2) Employing widely used and thoroughly tested attack implementations for defense evaluation;
> >
> >  (3) Ensuring the defense either restricts its threat model to scenarios it can reliably address or undergoes comprehensive evaluation against a wide range of possible attacks within the defined threat model;
> >
> >  (4) including comprehensive common benchmark tests to address potential side effects;
> >
> >  (5) exercising caution when claiming "unlearning", and some suggested tests for verifying unlearning claims.
> >
> >
> > We also note that this checklist is by no means exhaustive. The problem of evaluating defenses for open-weight LLMs is hard and is still at an early stage of research. We hope our checklist can be a starting point for future work to rule out at least some basic mistakes in evaluation.

---

> > > ### Author Response · Authors · 2024-11-26
> > > **Looking forward to discussing with the reviewer**
> > >
> > > Dear reviewer, we hope that our rebuttal has addressed your concerns or questions. If any aspects of our rebuttal remain unclear or require further clarification, please feel free to let us know. We would be happy to provide additional information or elaborate as needed. Thanks!

---

> > > > ### Comment · Reviewer_nnB1 · 2024-11-30
> > > >
> > > > I thank the authors for their detailed response and appreciate the clarifications on my questions. I have read other reviewers' reviews and I believe my concerns are addressed. I'm willing to revise my score to 6.

---

> > > > > ### Author Response · Authors · 2024-12-03
> > > > > **Thanks the reviewer**
> > > > >
> > > > > We sincerely thank the reviewer for their thoughtful and constructive engagement with our work. We are delighted to hear that our revisions and clarifications have addressed the reviewer’s questions and concerns. We also appreciate the reviewer for raising the score and the acknowledgment of our contributions.

---

### Official Review · Reviewer_KewE · 2024-11-02

**Soundness:** 3
**Presentation:** 3
**Contribution:** 3
**Rating:** 6
**Confidence:** 4

**Summary:**

The paper studies some weaknesses of two unlearning methods, RepNoise and TAR, that claim to be robust to attacks that can bring the knowledge back into the unlearned models.

Both methods and respective threat models are described in detail, together with pitfalls that break RepNoise and TAR defenses, such as:
- Trying different random seeds while keeping the rest of the attack unchanged
- Using the standard Huggingface SFT trainer instead of a custom trainer
- Small differences in fine-tuning configurations
- Different metric and evaluations produce completely different results

The authors conclude by explaining the difficulty of the problem and caution about claims of robustness.

**Strengths:**

- The paper is clear and well written, and has a thorough analysis of two unlearning methods and their weaknesses. The authors try very simple modifications that seem effective in breaking the safeguards. This is important as suggests to be very careful in the evaluation of unlearning methods.
- The authors include a discussion that explains how difficult the evaluation problem is, especially with the huge number of attacks available.

**Weaknesses:**

- It’s a standard practice to evaluate models on multiple-choice datasets by checking the highest logits of the letters corresponding to the answer options. So evaluating on a full answer by using humans and LLM judges sounds a bit of an unfair comparison. I agree that the practice of evaluating the logits instead of the full answer might be insufficient to properly compute the performance, but this sounds more a problem of the standard evaluation setting than the problem itself. Probably a more fair comparison would be to check the logits after adding the chat template.
- The discussion on lessons learned from the experiments is good, but it’d be nice to have, if possible, a summary of some evaluation principles that the authors retain to be relevant.

**Questions:**

- In evaluating other multiple-choice datasets, did you use human evaluation + LLM judge as well?
- What about other benchmarks? I’m checking Table 2 and it looks concerning that for some exact match benchmarks the results are zero.
- Are there other pitfalls you think might arise in the setting of robust unlearning?

---

> ### Author Response · Authors · 2024-11-22
> **Rebuttal - Part I**
>
> We are delighted that the reviewer finds our paper clear and well-written, and the problem we illustrate in the paper is important. We also appreciate the reviewer's constructive feedback and suggestions for improvement. We hope our following revision and clarifications can address the reviewer’s remaining concerns:
>
>
> **1. Clarifications regarding our evaluation setups for multiple-choice datasets**
> > It’s a standard practice to evaluate models on multiple-choice datasets by checking the highest logits of the letters corresponding to the answer options. So evaluating on a full answer by using humans and LLM judges sounds a bit of an unfair comparison. I agree that the practice of evaluating the logits instead of the full answer might be insufficient to properly compute the performance, but this sounds more a problem of the standard evaluation setting than the problem itself. Probably a more fair comparison would be to check the logits after adding the chat template.
>
>
> We agree with the reviewer that it's very common to evaluate models on multiple-choice datasets by `checking the highest logits of the letters corresponding to the answer options`. This approach is appropriate for utility performance tasks, such as those in the MMLU benchmark. High accuracy under this setup is a meaningful metric correlated with the model's task capability, making it a suitable basis for fair model-to-model comparisons.
>
>
> However, in our paper, for the evaluation case where we use `a full answer + humans and LLM judges`, the context is very different --- **our goal here is not to compare model capabilities fairly in a multiple-choice question (MCQ) task but rather to challenge whether the standard MCQ evaluation setup is even a reliable way to measure a model's unlearning of any specific knowledge.** More specifically, The rationale here is as follows:
>
>
>  * Here, we evaluate models on the WMDP benchmark [1], which is to measure whether the model has unlearned certain categories of weaponization knowledge. The promise of this benchmark is that --- if a model has unlearned the weaponization knowledge, it should not be able to answer the multiple-choice questions (MCQ) in the benchmark (consisting of weaponization knowledge questions) correctly, and the accuracy should be low. The defense such as TAR studied in this paper shows that the model with the safeguard indeed has a low accuracy on this benchmark and therefore claims the effectiveness of the defense.
>
>
>  * The challenge we raise is --- is a low accuracy on the WMDP benchmark under a certain evaluation setup (e.g., checking the highest logits of the letters corresponding to the answer options) really a reliable indicator that the model has unlearned the weaponization knowledge? We show that it is not necessarily the case --- by simply adding a prompt template and evaluating the answer in free-form text, the model's performance can be significantly better on this weaponization knowledge benchmark, and the model definitely still retains the weaponization knowledge to correctly answer the questions. Thus, a low accuracy on WMDP under the standard MCQ evaluation setup does not necessarily mean the model has unlearned the weaponization knowledge.
>
>
> Regarding the reviewer’s suggestion that "a more fair comparison would be to check the logits after adding the chat template": we agree and have already included such an evaluation in Figure 4. However, as shown, even with a chat template, the logits-based evaluation still indicates random accuracy on the WMDP benchmark, incorrectly suggesting the model may have unlearned the knowledge. In contrast, free-form text responses evaluated by human and LLM judges reveal higher accuracy, countering this conclusion.
>
>
> [1] Li, Nathaniel, et al. "The wmdp benchmark: Measuring and reducing malicious use with unlearning." arXiv preprint arXiv:2403.03218 (2024).
>
>
>
>
> > In evaluating other multiple-choice datasets, did you use human evaluation + LLM judge as well?
>
>
> No, for any other evaluation of multiple-choice datasets (except for the evaluation in Figure 4), we just follow the standard evaluation setup. This ensures that our reproduced studies follow the same setup as the original studies.
>
>
> Again, the use of the full answer + human evaluation / LLM judge setup in Figure 4 solely demonstrates that the standard MCQ evaluation is not always reliable for assessing unlearning.

---

> > ### Author Response · Authors · 2024-11-22
> > **Rebuttal - Part II**
> >
> > **2. A more organized summary of evaluation principles.**
> > > The discussion on lessons learned from the experiments is good, but it’d be nice to have, if possible, a summary of some evaluation principles that the authors retain to be relevant.
> >
> > We thank the reviewer for this suggestion. In our rebuttal revision, we added Appendix A to provide a more explicit and actionable checklist of items to consider for future evaluations of defenses to avoid similar pitfalls we identified in our paper. The checklist includes tips for:
> >
> >
> >  (1) Checking the robustness of the defense against attacks with different random seeds;
> >
> >
> >  (2) Employing widely used and thoroughly tested attack implementations for defense evaluation;
> >
> >  (3) Ensuring the defense either restricts its threat model to scenarios it can reliably address or undergoes comprehensive evaluation against a wide range of possible attacks within the defined threat model;
> >
> >  (4) including comprehensive common benchmark tests to address potential side effects;
> >
> >  (5) exercising caution when claiming "unlearning", and some suggested tests for verifying unlearning claims.
> >
> >
> > We also note that this checklist is by no means exhaustive. The problem of evaluating defenses for open-weight LLMs is hard and is still at an early stage of research. We hope our checklist can be a starting point for future work to rule out at least some basic mistakes in evaluation.
> >
> >
> >
> >
> > **3. Zero accuracy in Table 2**
> > > What about other benchmarks? I’m checking Table 2, and it looks concerning that for some exact match benchmarks, the results are zero.
> >
> >
> > **For all utility benchmarks, we follow their official standard evaluation pipeline** (See Appendix E3.2 for more details). All the utility tasks do not involve human evaluation, and we only use LLM judge for MT-Bench and TruthfulQA (following the official evaluation pipeline).
> >
> >
> > Indeed, as the reviewer also observed, the TAR checkpoint scores nearly zero on benchmarks like GSM8K, BBH, MATH, and HumanEval. That's exactly the point we want to highlight in this paper! The original TAR paper reported utility evaluation results solely on MMLU and MT-Bench, where the utility trade-offs appeared relatively acceptable. However, when we extended the evaluation to additional benchmarks in Table 2, we observed more severe performance degradation, with accuracy dropping to zero on some tasks.
> >
> >
> > We find that the reason why the TAR model has zero accuracy on these benchmarks is that the model suffers from some mode collapse on these benchmarks --- we have provided some qualitative examples in Appendix G, showing that the TAR model often generates nonsensical outputs for questions in these benchmarks.
> >
> >
> > By presenting these findings, we wish to underscore the challenges of fully capturing the side effects of defense mechanisms during evaluation. These results highlight the necessity for significant effort to achieve comprehensive coverage of diverse evaluation benchmarks. Our paper hopes to provide best practices so that future works thoroughly evaluate defenses and identify these side effects.
> >
> >
> > **4. Other pitfalls in evaluating unlearning**
> >
> >
> > > Are there other pitfalls you think might arise in the setting of robust unlearning?
> >
> >
> > Another possible pitfall that we recently noticed is that --- unlearning certain undesirable knowledge domains could sometimes also overly kill the model's performance on some relevant but completely benign tasks.  For example, when a model is trained to unlearn knowledge about bioweaponization, we find the model can sometimes fail to even answer basic benign biology questions such as "What is microbiology?".
> > A similar phenomenon is now called exaggerated safety in refusal-training-based alignment [1]. However, this problem has not been addressed well in the current unlearning benchmarks such as WMDP.
> >
> >
> > [1] Röttger, Paul, et al. "Xstest: A test suite for identifying exaggerated safety behaviours in large language models." arXiv preprint arXiv:2308.01263 (2023).

---

> > > ### Comment · Reviewer_KewE · 2024-11-25
> > > **Thank you for clarifying**
> > >
> > > Thank you for considering adding a list of items to consider for future evaluations.
> > >
> > > Thanks also for clarifying the other points. The drop in performance discussed at point 3 is quite surprising. A small decrease can be acceptable, but it's concerning the the scores after TAR are so low in at least 4 benchmarks. It makes sense to think that the model is quite damaged and can't be really used for certain tasks.
> > >
> > > Thank you.

---

> > ### Comment · Reviewer_KewE · 2024-11-25
> > **Follow-up**
> >
> > Thank you for clarifying, this is an important point (not just regarding unlearning, but more generally about how to evaluate models on different types on knowledge).
> >
> > Just a follow-up question on that. Using open-ended generation might provide better signal on the knowledge that a model has, however this can be hard to evaluate practically. For example, an LLM judge evaluating for biosecurity knowledge needs to know very well the topic in order to be able to classify whether an AI output is nonsensical or correct, and in addition might not be reliable anyway. And having humans judging AI outputs doesn't scale well and can be prohibitively costly, especially because one needs experts in a field to judge properly. Do you think there might be a good "compromise" between full MCQ evaluation and full open-ended?

---

> ### Author Response · Authors · 2024-11-26
> **Response to the reviewer's follow-up question**
>
> We sincerely thank the reviewer for the thoughtful engagement with our work. We are pleased to know that our revisions and clarifications have addressed the reviewer’s initial questions.
>
> Regarding the model’s performance drop after TAR, the reviewer’s observations are correct. Indeed, our independent evaluation shows that the model’s performance on multiple benchmarks can drop to concerningly low levels post-TAR. This is an important finding we highlight in our paper. Our goal is to caution researchers in this domain about such potential side effects and call for comprehensive evaluations to identify or rule out such side effects. This can make any proposed mitigation approach more convincing and reliable to use.
>
> **Regarding the reviewer’s follow-up question:**
> > Using open-ended generation might provide better signal on the knowledge that a model has, however this can be hard to evaluate practically. For example, an LLM judge evaluating for biosecurity knowledge needs to know very well the topic in order to be able to classify whether an AI output is nonsensical or correct, and in addition might not be reliable anyway. And having humans judging AI outputs doesn't scale well and can be prohibitively costly, especially because one needs experts in a field to judge properly. Do you think there might be a good "compromise" between full MCQ evaluation and full open-ended?
>
> The reviewer raises an excellent point regarding the trade-offs between open-ended-generation-based evaluation and the standard first-token-logits-based evaluation. We find this issue is moderate in the specific context we deal with in our paper. We clarify this point below.
>
> In our paper, we are primarily only working with the MCQ questions for the unlearning evaluation — the WMDP benchmark is a set of MCQ questions. By “open-ended generation”, we essentially mean that we allow the model to output its choice among (A,B,C,D) in an open-ended generation as opposed to relying on the logits of the first output token to decide the model’s answer. In this context, the LLM judge is only used to extract the model’s selected choice (among A~D) from the generated output. This is a simple task for the LLM judge model — the judge is only tasked with parsing the model’s output to extract its MCQ answer instead of using any deep domain expertise (e.g., biosecurity knowledge) to assess the correctness of the answer. We hope this clarification addresses the reviewer’s concerns and demonstrates how our approach navigates the trade-offs between practical feasibility and rigorous evaluation.
>
> More broadly, we agree with the reviewer’s point that scaling evaluations for open-ended generation tasks beyond MCQ-style questions is hard. It remains an ongoing challenge in the general LLM evaluation landscape, not just in the context of unlearning evaluation.

---

### Official Review · Reviewer_qb49 · 2024-11-02

**Soundness:** 3
**Presentation:** 3
**Contribution:** 4
**Rating:** 8
**Confidence:** 4

**Summary:**

The paper conducts multiple case studies to investigate the robustness of evals for unlearning harmful information from LLMs. They replicate published work with small changes to their evaluation methodology (i.e. the attack testing the defence) and show that this can nullify or even reverse the original findings.

**Strengths:**

S1: The paper is clearly motivated in regards to the importance of designing durable safeguards that ensure concepts are unlearned under adversarial settings.

S2: Replication studies are of high importance. Properly validating proposed methods has arguably higher value for the community than proposing yet another method.

S3: The authors are detailed-oriented in validating their replication study. For instance, they compare their exact replication with the original paper.

**Weaknesses:**

W1: There are various ambiguities in this work, that would benefit from more precision in language and technical details provided.

- W1a: This work uses ambiguous language for technical concepts at times. Formalising some concepts using math could help to greately disambiguate concepts that are vague in text. For instance, L136 "RepNoise trains a model to push its representations of HarmfulQA data points at each layer toward random noise." For readers not familiar with prior work, this statement is hardly comprehensible. Formalising objectives can help to clarify.

- W1b: The paper is sometimes a too ambiguous in language when presenting results. For instance, "the model with RepNoise does not respond to HarmfulQA questions more than a moderate amount" (L138). What is a moderate amount?

- W1c: The specific details of attacks and threat models are not always entirely clear. For instance, "An attacker attempts to recover this knowledge through fine-tuning." (L163) is very vague. What data is this fine-tuning performed on? Fine-tuning on a high quality dataset of weaponization knowledge will _always_ improve performance and unlearning cannot prevent that.

W2: The paper could be more explicit about the lessons learned from failing to replicate results reliably. The findings could be more _actionable_. For instance, providing a checklist for future evaluations might be useful.

W3: The hyperparameters of the replication study and original study can be a bit hard to follow at times. This is true for the main body and Appendix alike. More tabular overviews like Table 1 would be highly appreciated.

W4: At this point I cannot verify the claims of the reproducibility statement ("our source code is included in our supplementary materials") as I do not have access to the supplementary material.

_Minor:_

MW1: L116: "This is a standard practice - " comes a little sudden.

MW2: Figure 6: Figures placed at the top or bottom of the page are generally preferred by the community as they make it easier to read the text and distinguish between caption and main body.

**Questions:**

Q1: L230: Do the authors use use greedy decoding in this work exactly like the original paper?

Q2: The authors run 5 seeds for most of their experiments. 4 new seeds + the original seed from prior work?

Q3: Huggingface is a super high-level framework that obfsucates a lot of details that are hard to grasp and alter training in the most unexpected ways. What did the authors do to ensure that the HF SFT Trainer does not violate some assumption of the threat model? What are the exact differences in how the HF trainer works in comparison to the original codebases?

Q4: How are the datasets used for the attack fine-tuning set up exactly? One can take the most harmless model and fine-tune it on dangerous datasets and it will respond to harmful questions.

Q5: Figure 4 does not say what error bars are. What is each bar aggregated over?

Q6: The authors describe benchmarks that they were not able to replicate (interesting finding). They do not talk about experiments they were able to replicate robustly (null finding in this context).

---

> ### Author Response · Authors · 2024-11-22
> **Rebuttal - Part I**
>
> We are delighted that the reviewer finds our work clearly motivated and our replication studies detail-oriented and of high importance. We also thank the reviewer for the constructive feedback and suggestions for improvement. We hope our following revision and clarifications can address the reviewer’s remaining concerns:
>
>
>
>
> **1. We have revised our paper to address several ambiguities pointed out by the reviewer in our presentation.**
>
>
> > W1a: This work uses ambiguous language for technical concepts at times. Formalising some concepts using math could help to greately disambiguate concepts that are vague in text. For instance, L136 "RepNoise trains a model to push its representations of HarmfulQA data points at each layer toward random noise." For readers not familiar with prior work, this statement is hardly comprehensible. Formalising objectives can help to clarify.
>
>
> Following the reviewer's suggestion, we have added a new Appendix C to provide a more formal introduction to RepNoise and TAR. There, we have mathematically formalized the training objectives of the two approaches for better clarity and precision. We also added reference links to the main text of this appendix, where these concepts are introduced.
>
>
> > W1b: The paper is sometimes a too ambiguous in language when presenting results. For instance, "the model with RepNoise does not respond to HarmfulQA questions more than a moderate amount" (L138). What is a moderate amount?
>
>
> Following the reviewer's suggestions, we have tightened our language here to avoid ambiguity. We revised the initial sentence `the model with RepNoise does not respond to HarmfulQA questions more than a moderate amount` to `the model with RepNoise can still consistently refuse over 90% of HarmfulQA questions from the test set.`
>
>
>
>
> > W1c: The specific details of attacks and threat models are not always entirely clear. For instance, "An attacker attempts to recover this knowledge through fine-tuning." (L163) is very vague. What data is this fine-tuning performed on? Fine-tuning on a high quality dataset of weaponization knowledge will always improve performance and unlearning cannot prevent that.
>
>
> As the reviewer also pointed out, for the threat model of TAR, there isn't a clear restriction on what data the attack can use for fine-tuning. We would like to clarify that this is not an ambiguity in our writing but rather an accurate representation of the strong and broad threat model faced by TAR. To clarify, TAR is designed to introduce durable safeguards into open-weight LLMs, ensuring that adversaries cannot recover weaponization knowledge even after fine-tuning the model for several thousand steps. Our critique in the paper emphasizes that this threat model may be overly strong and broad; evaluating defenses against fine-tuning attacks on a single dataset is insufficient to assess their overall effectiveness.
>
> **2. We have added a checklist of evaluation pitfalls to make our learned lessons more explicit and actionable.**
>
>
> > W2: The paper could be more explicit about the lessons learned from failing to replicate results reliably. The findings could be more actionable. For instance, providing a checklist for future evaluations might be useful.
>
>
>
>
> We thank the reviewer for this suggestion. In our rebuttal revision, we added Appendix A to provide a more explicit and actionable checklist of items to consider for future evaluations of defenses to avoid similar pitfalls we identified in our paper. The checklist includes tips for:
>
>
>  (1) Checking the robustness of the defense against attacks with different random seeds;
>
>
>  (2) Employing widely used and thoroughly tested attack implementations for defense evaluation;
>
>  (3) Ensuring the defense either restricts its threat model to scenarios it can reliably address or undergoes comprehensive evaluation against a wide range of possible attacks within the defined threat model;
>
>  (4) including comprehensive common benchmark tests to address potential side effects;
>
>  (5) exercising caution when claiming "unlearning", and some suggested tests for verifying unlearning claims.
>
>
> We also note that this checklist is by no means exhaustive. The problem of evaluating defenses for open-weight LLMs is hard and is still at an early stage of research. We hope our checklist can be a starting point for future work to rule out at least some basic mistakes in evaluation.

---

> > ### Author Response · Authors · 2024-11-22
> > **Rebuttal - Part II**
> >
> > **3. We have improved the clarity of our presentation for the hyperparameters of the replication study.**
> >
> >
> > > W3: The hyperparameters of the replication study and original study can be a bit hard to follow at times. This is true for the main body and Appendix alike. More tabular overviews like Table 1 would be highly appreciated.
> >
> >
> > We appreciate the reviewer’s thoughtful feedback. We added a new Table 3 (See L957-L962) to better introduce the hyperparameter configurations used in RepNoise. For the hyperparameter configurations used by TAR, we detailed them in Table1 and Table 5.
> >
> >
> >
> >
> > **4. Supplementary materials**
> >
> >
> > > At this point I cannot verify the claims of the reproducibility statement ("our source code is included in our supplementary materials") as I do not have access to the supplementary material.
> >
> >
> > We have provided the source code in our supplementary materials by the time of our initial submission. If the reviewer has any difficulty accessing the supplementary materials, we will be happy to assist the reviewer in addressing any issues the reviewer meets.
> >
> >
> >
> >
> > **5. Other minor presentation issues**
> >
> >
> > We thank the reviewer for the super careful review. We will take into account the reviewer's suggestions, and continue to improve the clarity and readability of our paper for our final camera-ready version.

---

> > > ### Author Response · Authors · 2024-11-22
> > > **Rebuttal - Part III**
> > >
> > > **6. Answers to the reviewer's other questions**
> > >
> > >
> > >
> > >
> > >
> > >
> > > >Do the authors use greedy decoding in this work exactly like the original paper?
> > >
> > >
> > > A: As mentioned in L1037, we do sampling with temperature=0.9, top_p=0.6, max_tokens=2048 in our re-evaluation with our own codebase. When reproducing the results using the official codebase, we used the same sampling strategy as the original paper.
> > >
> > >
> > >
> > >
> > > > The authors run 5 seeds for most of their experiments. 4 new seeds + the original seed from prior work?
> > >
> > >
> > > A: For the data sampling process of the original codebase from both Repnoise and TAR, they use the sequential sampler instead of the random sampler. Therefore, in terms of the data generation process, random seeds won’t change the result. As mentioned in L230, L966, and L1060, in our re-evaluation using the official codebase, we changed the sampler into the random sampler and randomly selected 5 seeds.
> > >
> > >
> > >
> > >
> > > > Huggingface is a super high-level framework that obfsucates a lot of details that are hard to grasp and alter training in the most unexpected ways. What did the authors do to ensure that the HF SFT Trainer does not violate some assumption of the threat model? What are the exact differences in how the HF trainer works in comparison to the original codebases?
> > >
> > >
> > > A: We can confirm that the HF SFT Trainer does not violate the threat model assumptions. In the threat models considered in this paper, the adversary is allowed to arbitrarily fine-tune the model's weights. The HF SFT Trainer implements a standard supervised fine-tuning algorithm, which falls strictly within the scope of the defined threat model.
> > >
> > >
> > > Identifying the exact differences between the HF trainer and the original codebases is indeed challenging, as the reviewer noted. HF is a high-level framework that abstracts away many low-level details, which can be difficult to trace. This abstraction is one of the underlying reasons why we recommend in Appendix A that future work evaluating defenses against fine-tuning attacks should, by default, consider using broadly tested and widely adopted fine-tuning implementations (e.g., the HF implementation). These implementations benefit from years of optimization efforts, and their use can mitigate the influence of potentially overlooked low-level details on the evaluation results. By leveraging well-established implementations, researchers can reduce the evaluation burden and ensure consistency across studies.
> > >
> > >
> > >
> > >
> > >
> > >
> > >
> > >
> > > > How are the datasets used for the attack fine-tuning set up exactly? One can take the most harmless model and fine-tune it on dangerous datasets and it will respond to harmful questions.
> > >
> > >
> > > A: We use the exact same datasets from RepNoise (a subset of Beavertails-30k-Train) and TAR (for the forget set we use a subset of Pile-Bio forget, for the retain set we use a mixture of Magpie-Align and Pile-bio retain, see L1063-L1068) for fine-tuning attack evaluation. For benign fine-tuning on RepNoise, we use AOA and Alpaca-Salient as our datasets (See L1008-L1014)
> > >
> > >
> > > Also, the reviewer mentioned, `one can take the most harmless model and fine-tune it on dangerous datasets, and it will respond to harmful questions.`. We want to clarify that this is exactly the problem that durable safeguards aim to address. For open-weight LLMs, the goal of RepNoise and TAR considered in this paper is to build durable safeguards in the models such that even if the adversary fine-tunes the model on harmful datasets, the model should still behave safely. Both RepNoise and TAR claim that this is a feasible problem, and they can make models safe even against harmful fine-tuning on harmful datasets. The goal of this paper is to verify such claims and also illustrate the challenges in evaluating these claims.
> > >
> > >
> > >
> > >
> > > > Figure 4 does not say what error bars are. What is each bar aggregated over?
> > >
> > >
> > > A: The error bar is aggregated over 3 repetitions with different random seeds(Due to the high cost of human evaluation, we did not repeat this experiment 5 times). We added these details to the caption in our revised version.
> > >
> > >
> > >
> > >
> > > > The authors describe benchmarks that they were not able to replicate (interesting finding). They do not talk about experiments they were able to replicate robustly (null finding in this context).
> > >
> > >
> > > A: We can robustly replicate the MMLU score and the WMDP score of the TAR-checkpoint before fine-tuning attack. Because when evaluating these benchmarks using the logits order of the first token, there’s no randomness in the whole process. Therefore, we can robustly replicate these results.

---

> > ### Comment · Reviewer_qb49 · 2024-11-25
> >
> > I thank the authors for their detailed response. I appreciate the clarifications on my questions, the checklist and the Appendix C. I also managed to locate the supplementary materials.
> >
> > I maintain a very positive view on this paper and believe it makes valuable contributions.

---

> > > ### Author Response · Authors · 2024-11-26
> > > **Thanks the reviewer**
> > >
> > > We sincerely thank the reviewer for the thoughtful and constructive engagement with our work. We are delighted to hear that our revisions and clarifications have addressed the reviewer’s questions. We also appreciate the reviewer’s positive assessment of our paper and the acknowledgment of our contributions.

---

### Official Review · Reviewer_2sdp · 2024-11-04

**Soundness:** 3
**Presentation:** 3
**Contribution:** 3
**Rating:** 6
**Confidence:** 4

**Summary:**

The paper investigates the durability of safeguards for open-weight large language models (LLMs) in the face of adversarial fine-tuning. It critiques current methods, focusing on two specific approaches: Representation Noising (RepNoise) and Tamper Attack Resistance (TAR). Through multiple case studies, the authors show that these defenses are fragile, with minor adjustments in evaluation or attack settings often bypassing the safeguards. They also emphasize the importance of clear and constrained threat models to avoid misleading security claims, especially given the high variance in attack success based on random seeds, prompt formats, and hyperparameter choices.

**Strengths:**

1.Relevance to Security: The paper addresses an important and timely issue—LLM security in open-weight contexts—by examining whether current safeguards are genuinely robust.

2.Empirical Rigor: The study is thorough, covering multiple aspects of how small changes in setup, configuration, and prompting can influence the success of fine-tuning attacks.

3.Practical Implications: The paper offers valuable insights for researchers and policymakers aiming to secure LLMs, highlighting the limitations of existing approaches in durable defense.

**Weaknesses:**

1.Lack of Alternative Solutions: While the paper critiques existing methods, it lacks exploration of alternative approaches or improvements that could enhance durability.

2.Heavy Reliance on Specific Models and Datasets: The findings are heavily based on specific models and datasets (e.g., LLaMA-2, WMDP), potentially limiting generalizability.

3.High Computational Costs: The methodology requires considerable computational resources for testing different random seeds, prompts, and configurations, which may limit its scalability.

**Questions:**

1.Could the authors provide insights on how future safeguards might address the identified weaknesses without significantly increasing computational requirements?

2.How does the approach generalize to different model architectures or datasets beyond the ones used in this study?

---

> ### Author Response · Authors · 2024-11-22
> **Rebuttal - Part I**
>
> We appreciate the reviewer's recognition of the importance of the issue addressed in our paper, as well as the acknowledgment of the empirical rigor and the valuable practical implications of our work. We hope our following clarifications can address the reviewer’s remaining concerns:
>
> **1. Exploration of Alternative Solutions**
>
> We thank the reviewer for bringing up this issue. Below is our clarification on this point:
> * **The point of this paper is not to attempt to solve the problem but to establish baseline evaluation principles for any future work that attempts to make progress in this problem.** A painful lesson we learned from AI security research over the last decade is that many defenses failed to withstand rigorous scrutiny due to flawed or incomplete evaluations (e.g., [1,2,3]). This highlights the importance of establishing standard baseline evaluation principles for evaluating defenses, as exemplified by [4,5]. Only if we first address the meta-problem of evaluating defenses can we genuinely track progress in building defenses and confidently draw security conclusions.
>     However, we notice this evaluation issue has not received sufficient attention in recent work (such as RepNoise and TAR, which we examined in this paper) that attempts to produce durable safeguards for open-weight LLMs. These studies initially made strong claims about the effectiveness of their defenses, but upon re-evaluation, we found that they can be easily bypassed and that their evaluations contain various pitfalls, leading to overestimation of their defenses' effectiveness. This motivated us to conduct our study. By using these two recent defenses as case studies, we illustrate and highlight common evaluation pitfalls in this problem area, and we hope that our findings can serve as important evaluation principles for future work attempting to make progress in this problem.
> * **Why didn't we explore alternative solutions?** The reviewer is correct that this work did not explore alternative approaches for durably safeguarding open-weight LLMs. We didn't do so because safeguarding an open-weight model when the adversary can arbitrarily modify the model's weights is a difficult open problem. The goal of our paper is not to develop a new solution, but to ensure that any exploration of solutions to this challenging problem properly evaluates the solutions and avoids pitfalls.
>     To appreciate its difficulty, consider that so far, we do not even have a solution for adversarial examples, where the adversary only "fine-tunes" the input to the model --- and it is already 2024, we still see researchers try very hard to solve the problem but the solution was then quickly shown to fail in openreview comments (https://openreview.net/forum?id=IHRQif8VQC).
>      Now, we have a much harder problem: for open-weight LLMs, how can we defend against a much stronger adversary who can directly "fine-tune" the model's weights, not just the input, as in adversarial examples? It is even unclear under what conditions durably safeguarding open-weight models is a feasible goal to pursue under such a strong threat model. All these remain open research questions to date.
> * **Another point of this paper is to inform practitioners of the difficulty of safeguarding open-weight LLMs with current technologies.** Not all practitioners in AI safety and security are fully aware of the actual effectiveness of current defense approaches. Our study provides solid evaluation evidence to help stakeholders calibrate their expectations regarding the feasibility of durably safeguarding open-weight LLMs with existing technologies.
> For example, a recent public interest comment (https://www.mercatus.org/research/public-interest-comments/comment-nist/aisi-ai-800-1-expanding-guidelines-open-source-ai) suggested that the U.S. National Institute of Standards and Technology (NIST) include approaches like TAR in their guidelines for managing open-source foundation model misuse risks. Our evaluation offers an evidence-based counterpoint, indicating that users and policymakers should be cautious about relying on these approaches in their current state.
>
> [1] Carlini, Nicholas, and David Wagner. "Defensive distillation is not robust to adversarial examples." arXiv preprint arXiv:1607.04311 (2016).
>
> [2] Carlini, Nicholas, and David Wagner. "Magnet and" efficient defenses against adversarial attacks" are not robust to adversarial examples." arXiv preprint arXiv:1711.08478 (2017).
>
> [3] Athalye, Anish, Nicholas Carlini, and David Wagner. "Obfuscated gradients give a false sense of security: Circumventing defenses to adversarial examples." International conference on machine learning. PMLR, 2018.
>
> [4] Carlini, Nicholas, et al. "On evaluating adversarial robustness." arXiv preprint arXiv:1902.06705 (2019).
>
> [5] Tramer, Florian, et al. "On adaptive attacks to adversarial example defenses." Advances in neural information processing systems 33 (2020): 1633-1645.

---

> > ### Author Response · Authors · 2024-11-22
> > **Rebuttal - Part II**
> >
> > **2. Generalizability of our study**
> > > The findings are heavily based on specific models and datasets (e.g., LLaMA-2, WMDP), potentially limiting generalizability.
> >
> > Our experiments in this paper are to show how the security evaluations in existing work for durably safeguarding open-weight LLMs are flawed and lead to overestimation of the defenses' effectiveness. We chose the specific models or datasets in our experiments because these defenses utilize them to showcase their effectiveness.
> >
> > Moreover, the evaluation pitfalls and principles identified in our work are not tied to any particular model or dataset—they are consolidated to be generic and widely applicable. For example, our discussion of randomness in evaluation (Section 3.1), the implementation details of fine-tuning attack trainers (Section 3.2), the adaptiveness of fine-tuning attack configurations (Section 3.3), and the choices of prompt templates and metrics (Section 3.4) apply universally, regardless of the specific model or dataset being evaluated.
> >
> > Similarly, the principles we outline for assessing whether unlearning methods genuinely remove targeted information, as well as the guidelines for reporting evaluation results and drawing security conclusions, are also model-agnostic and broadly applicable across defense evaluation scenarios (Section 4).
> >
> >
> > **3. Computational Costs**
> > > High Computational Costs: The methodology requires considerable computational resources for testing different random seeds, prompts, and configurations, which may limit its scalability.
> >
> > We do agree with the reviewer that covering all these testing items we suggest in our paper requires considerably more computational resources. This is also one of the reasons why we state that even the evaluation of defenses for open-weight LLM safeguarding is hard.
> >
> > That said, for any trustworthy defense under such a strong threat model, these testing items constitute only the minimal baseline requirements. Security is fundamentally about minimizing vulnerabilities, and rigorous testing is a non-negotiable step to ensure the reliability and robustness of any proposed defense. Therefore, the narrative here should not center on reducing evaluation costs for defense developers but rather on advocating for increased investment of effort and resources to guarantee that defenses are comprehensively and rigorously evaluated before practical deployment.

---

> > > ### Author Response · Authors · 2024-11-26
> > > **Looking forward to discussing with the reviewer**
> > >
> > > Dear reviewer, we hope that our rebuttal has addressed your concerns or questions. If any aspects of our rebuttal remain unclear or require further clarification, please feel free to let us know. We would be happy to provide additional information or elaborate as needed. Thanks!

---

> > > > ### Comment · Reviewer_2sdp · 2024-11-28
> > > > **Considerations for Score Adjustment**
> > > >
> > > > Furthermore, Table 2 of the paper mentions that defense measures may improve model performance on specific safety tasks while sacrificing performance on other tasks. The discussion of the extent to which defense measures lead to a decrease in model performance on other tasks is somewhat cursory. If such performance trade-offs are unavoidable, what level of performance loss does the author consider acceptable? What kind of trade-off should be considered indicative of an effective defense measure? Since the paper aims to propose a standard, I believe an analysis of this issue would better guide the assessment and balance of defense measures in terms of performance trade-offs.
> > > >
> > > > If the author provides a clear explanation of this issue, I would be willing to consider a score increase.

---

> > > > > ### Author Response · Authors · 2024-12-02
> > > > > **Response to the reviewer's follow-up comment - Part II**
> > > > >
> > > > > **Side effects and acceptable trade-off?** As noticed by the reviewer, one important finding in our paper is that a defense could bring side effects — sometimes tremendous (e.g., after applying TAR, the model’s accuracy drops to 0% on some utility benchmarks). The point we want to highlight is that — the evaluation can fail to comprehensively capture such side effects — for example, the TAR paper didn’t find many of the side effects we report in this paper. Therefore, **our recommended standard is that future work must adopt holistic evaluations across multiple utility benchmarks to provide a clearer picture of how defenses affect model performance on benign utility tasks.** Transparent reporting of these side effects will help stakeholders to make more informed decisions.
> > > > >
> > > > > As for the acceptable trade-off between defense efficacy and side effects, we do not have an objective and universal answer. The acceptable level of trade-off depends on the subjective preferences of stakeholders and the specific context in which the model is used. For applications where safety and security are of the highest priority, a higher tolerance for side effects may be acceptable. Conversely, in scenarios where model performance is critical, stakeholders may find such trade-offs less acceptable. However, even though there’s no exact threshold, the zero score in some utility benchmarks for TAR checkpoints is illustrative of undesirable mode collapse behaviour and should be investigated further (we also provided some qualitative examples in Appendix G). We hope the reviewer appreciates our perspective that we cannot have an exact recommendation for what is an acceptable trade-off in an evaluation checklist, but we do recommend a comprehensive evaluation across multiple utility benchmarks to holistically capture such side effects and transparently report them.

---

> > > > > > ### Comment · Reviewer_2sdp · 2024-12-03
> > > > > >
> > > > > > Thank you for your thoughtful response. I now have a clearer understanding of the paper’s goals and appreciate the emphasis on establishing rigorous evaluation standards for future defenses. The clarification on performance trade-offs and side effects, as well as the recommendation for holistic evaluations, addresses my concerns. I agree that while a concrete roadmap is challenging at this stage, the proposed checklist will significantly guide future research. Based on these clarifications, I am willing to revise my score to 6.

---

> > > > > > > ### Author Response · Authors · 2024-12-03
> > > > > > > **Thanks the reviewer**
> > > > > > >
> > > > > > > We sincerely thank the reviewer for their thoughtful and constructive engagement with our work. We are delighted to hear that our revisions and clarifications have addressed the reviewer’s questions and concerns. We also appreciate the reviewer for raising the score and the acknowledgment of our contributions.

---

> > > ### Comment · Reviewer_2sdp · 2024-11-28
> > > **Thanks for clarifying**
> > >
> > > Thank you for your response. Based on your further explanation, I now have a clearer understanding of the paper's objective, which is to establish baseline evaluation principles for any future work that aims to make progress in this area. Additionally, the inclusion of Appendix A, which provides a more explicit and actionable checklist of items to consider for future evaluations of defenses to avoid similar pitfalls identified in the paper, is a meaningful contribution.
> > >
> > > However, the paper primarily presents validation analysis experiments, and a systematic research roadmap and theoretical analysis are still lacking. This reduces the overall contribution of the work. Moreover, while the paper emphasizes the many challenges faced by current defense measures and criticizes the potential underestimation or misleading results of these defenses, without proposing a clear research roadmap or solutions, it risks being perceived as a "problem-oriented" rather than a "solution-oriented" discussion. This could leave readers feeling pessimistic and lacking confidence in the progress of current defense technologies. This may hinder me from giving a higher score.

---

> ### Author Response · Authors · 2024-12-02
> **Response to the reviewer's follow-up comment - Part I**
>
> We thank the reviewer for the thoughtful engagement and detailed feedback on our work. We are pleased to know that our revisions and clarifications have made the objective of this paper clearer. We also appreciate the reviewer’s recognition of the value of the actionable checklist we provide, which aims to guide future evaluations and address the pitfalls highlighted in the paper. We hope the following clarifications can help address the reviewer’s remaining concerns:
>
> **Research roadmap and pessimism in the progress.** We wish to acknowledge upfront that this paper indeed does not propose a practical roadmap for solving the problem of durably safeguarding open-weight LLMs. As discussed in our earlier rebuttal, this is because this problem remains a fundamentally hard and open problem. Particularly, this problem is expected to be more difficult than addressing adversarial examples — and it’s known that even for addressing adversarial examples, a clear roadmap remains absent despite over a decade of research efforts. So, we hope the reviewer can appreciate that developing such a roadmap for durably safeguarding open-weight LLMs is a highly non-trivial agenda, one that warrants a dedicated line of work separate from the scope of this paper. From a positive perspective, we do believe these defense mechanisms (including future methods evolving from RepNoise and TAR) could potentially *increase* the cost to adversaries, even if they don’t provide high worst-case security guarantees, **but evaluations should accurately and rigorously reflect the added costs for adversaries.** Otherwise, an incomprehensive evaluation may neglect the potential side effects brought by the defense. For instance, the original evaluation of RepNoise and TAR failed to capture the high variance of the model’s robustness against fine-tunining attack due to the lack of evaluating with different random seeds. A key contribution of our work is to provide constructive recommendations to avoid such biased evaluation in our paper.
>
> We also understand the reviewer’s concern that the paper may project a pessimistic view on the progress of developing defenses. However, we believe that this paper will have a positive impact on advancing the field. The baseline evaluation checklist we propose establishes a more rigorous protocol for assessing the correctness and effectiveness of defenses. This will enable future research to better track progress, avoid common pitfalls, and mitigate the risk of pursuing directions that yield seemingly promising results but are ultimately artifacts of misevaluation. This dynamic has been observed in the previous study of adversarial examples, where a series of critical adaptive attack studies [1,2,3,4,5] have played a pivotal role in identifying which defenses are effective and which are not.
>
> [1] Carlini, Nicholas, and David Wagner. "Defensive distillation is not robust to adversarial examples." arXiv preprint arXiv:1607.04311 (2016).
>
> [2] Carlini, Nicholas, and David Wagner. "Adversarial examples are not easily detected: Bypassing ten detection methods." Proceedings of the 10th ACM workshop on artificial intelligence and security. 2017.
>
> [3] Athalye, Anish, Nicholas Carlini, and David Wagner. "Obfuscated gradients give a false sense of security: Circumventing defenses to adversarial examples." International conference on machine learning. PMLR, 2018.
>
> [4] Carlini, Nicholas, et al. "On evaluating adversarial robustness." arXiv preprint arXiv:1902.06705 (2019).
>
> [5] Tramer, Florian, et al. "On adaptive attacks to adversarial example defenses." Advances in neural information processing systems 33 (2020): 1633-1645.

---

### Meta-Review · Area_Chair_WMB3 · 2024-12-13

**Metareview:**

The paper critiques the durability of safeguards for open-weight LLMs, focusing on two methods, RepNoise and TAR. It reveals that minor changes in attack setups can bypass these defenses, highlighting evaluation pitfalls and calling for stricter, more transparent protocols. While the paper is empirically thorough and provides practical insights, its scope is limited to two methods, and it does not propose alternative solutions. Despite these limitations, its contributions to improving evaluation standards are valuable.

**Additional Comments On Reviewer Discussion:**

During the rebuttal, reviewers raised concerns about clarity, lack of alternative solutions, narrow focus, and reproducibility. The authors addressed these by formalizing methods, improving clarity, adding an evaluation checklist, and emphasizing the general applicability of their findings. While the lack of solutions and narrow scope remain limitations, the improved rigor and actionable contributions outweigh these concerns, leading to a recommendation for acceptance

---

### Decision · Program_Chairs · 2025-01-22

Accept (Poster)